# Sesquiterpenoids from *Inula britannica* and Their Potential Effects against Triple-Negative Breast Cancer Cells

**DOI:** 10.3390/molecules27165230

**Published:** 2022-08-16

**Authors:** Ruo-Yu Qi, Cong Guo, Xiao-Na Peng, Jiang-Jiang Tang

**Affiliations:** Shaanxi Key Laboratory of Natural Products & Chemical Biology, College of Chemistry & Pharmacy, Northwest A&F University, Taicheng Road, Yangling 712100, China

**Keywords:** *Inula britannica*, sesquiterpenoid, triple-negative breast cancer, dimers

## Abstract

Flowers of *Inula britannica* commercially serve as pharmaceutical herbs in the manufacturing of medicinal products. In the current study, sesquiterpenoids of *I. britannica* flowers’ extract and their potential effects against triple-negative breast cancer (TNBC) cells were investigated. Eight structurally diverse sesquiterpenoids, including one sesquiterpenoid dimer (**1**) and seven sesquiterpenoid monomers (**2**–**8**) were isolated from this source. The structures of all compounds were elucidated by 1D/2D NMR data, and their absolute configurations were discerned by single crystal X-ray diffraction. All of the compounds were tested for their potential effects against TNBC. Specifically, **5** displayed strong antiproliferative potency against TNBC cells with a high selective index (SI) on MCF-7 cells (SI > 4 of IC_50_ on MDA-MB-468/IC_50_ on MCF-7), and dimer **1** (IC_50_ = 8.82 ± 0.85 μM) showed better antiproliferative potency against MCF-7 cells than the other monomers did (**2**–**8**) (IC_50_ > 20 μM). To our best knowledge, compound **5** is the first sesquiterpenoid targeting TNBC cells.

## 1. Introduction

The biggest cancer-related disease for women worldwide remains breast cancer (BrCa) [1,2]. BrCa subtypes are defined by their histopathological appearance and expression of estrogen receptors (ER), progesterone receptors (PR) and human epidermal growth factor receptor 2 (HER2). Most of the cases are ER-positive and PR-positive BrCa, and hormone therapy normally shows really good effects in them [3]. HER2-positive BrCa takes about 15–20% of all kinds of BrCa. Today, numerous targeted anti-HER2 medications have been authorized for use in treating HER2-positive BrCa [4]. Unfortunately, 10–20% of BrCa is known as triple-negative breast cancer (TNBC) and shows a negative result for ER, PR and HER2 expression. TNBC, with poor overall survival (OS) and an aggressive clinical course, is a poorly understood BrCa type. The incidence of TNBC is amazingly high among overweight [5], non-Hispanic black, and younger women [3,5,6].

Because of the antioxidant and anti-neuroinflammatory activities they have, natural-product-derived bioactive agents have attracted attention on the development of preventive neuroprotectants or nutraceuticals for the treatment of neurodegenerative disorders [7,8,9,10]. In Eastern Asia, the *Inula britannica* plant has been used to treat disorders of the digestive system, bronchitis and inflammation for many years [11,12]. Numerous biologically active sesquiterpene lactones have been identified from this plant, according to earlier phytochemical studies [13]. Several studies have shown that certain sesquiterpene lactones can prevent breast cancer cells from migrating and invading [14,15]. For example, eupatolide lowers inflammation by inhibiting the nuclear factor-light-chain-enhancer of activated B cells, as well as preventing the production of tumor necrosis factor (TNF) and NO [16,17,18]. Moreover, it is also reported that eupatolide could play an essential role in the sensitization of TRAIL-induced apoptosis in human breast cancer cells. Therefore, it is particularly important to explore novel bioactive metabolites from *I. britannica* flowers and their anti-tumor activities [13].

In our continuous efforts to explore natural products against TNBC, the *I. britannica* flowers was further investigated and isolated to obtain eight structurally diverse sesquiterpenoids, including one known sesquiterpenoid dimer (**1**) and seven known sesquiterpenoid monomers (**2**–**8**). The antiproliferative potential of the isolates was tested on breast cancer cell lines MDA-MB-231, MDA-MB-468 and MCF-7.

## 2. Results and Discussion

The dried flowers of *I. britanica* were pulverized and extracted with 95% aqueous ethanol, and the extract was successively partitioned with petroleum ether, CH_2_Cl_2_, and EtOAc. One known sesquiterpenoid dimer (**1**) and seven known sesquiterpenoid monomers (**2**–**8**) (Figure 1) were isolated from the column chromatography separation of the ethyl acetate layer. The eight compounds were identified as japonicone H (**1**) [19], 1β,4β-dihydroxy-8β-acetoxy-5αH-eudesma-11(13)-en-12-oic acid methyl ester (**2**) [20], 6β-hydroxytomentosin (**3**) [21], budlein B (**4**) [22,23], 1,10β-dihydroxy-4αH-1,10-secoeudesma-5(6),11(13)-dien-12,8β-olide (**5**) [18], 1β-hydroxy-11,13-dihydroisoalantolactone (**6**) [24], 6-O-isobutyrylbritannilactone (**7**) [25,26], (1R,4S,5R,7R,8R,10S)-1,5-dihydroxy-eudesma-11(13)-en-12,8-olide (**8**) [20]. The structures of all isolates were well-characterized by 1D/2D NMR analysis referred to in the Appendix A.

Structurally, compound **1** is an endo [4+2] Diels–Alder dimeric sesquiterpene lactone of 1,10-secoeudesmanolides moiety and a guaiane monomer, which has been firstly isolated from *Inula japonica* Thunb [17]. In addition, compound **6** was also isolated from this species for the first time. The X-ray crystallographic analysis (Figure 2) showed that there was hydrogen bonding between the C_8_-OH and C_15_-OH in **4** that has not been reported in a previous study [23]. We also optimized the flack parameter from −0.06 (**7**) to −0.04 (**5**), which allowed us to get a more specific explicit assignment of the absolute structure.

The potent anticancer properties of sesquiterpenoids have been reported to be chemically mediated by the active motifs α-methylene-γ-butyrolactone that all these isolates contain. The α-methylene-γ-lactone group can form a covalent adduct with important cellular protein targets, such as -SH residues [27]. To examine whether isolates **1**–**8** have antiproliferative effects on breast cancer cells, triple-negative breast cancer cell lines MDA-MB-231 and MDA-MB-468, and estrogen receptor-positive breast cancer cell line MCF-7 were inoculated into 96-well plates. After incubating for 24 h, the medium was replaced with a fresh one containing the specified compounds (0.1, 0.5, 1, 5, 10, 20 μM). After 48 h, the antiproliferative effects were determined using the SRB assay by measuring cellular protein content. Doxorubicin (DOX) was used as the positive control. The IC_50_ values are summarized in Table 1.

In the compound treatment periods, compound **1** showed antiproliferative activities against MDA-MB-468 and MCF-7 cells with IC_50_ of 6.68 ± 0.70 μM and 8.82 ± 0.85 μM. However, with MDA-MB-231 cells, compound **1** did not show much of the antiproliferative activities with IC_50_ > 20 μM. **1** as a sesquiterpenoid dimer showed better antiproliferative activities against MCF-7 breast cancer cells than all other seven sesquiterpenoid monomers. Compound **5** showed antiproliferative effects on MDA-MB-231 and MDA-MB-468 cells with IC_50_ of 11.5 ± 0.71 μM and 4.92 ± 0.65 μM. However, **5** had no significant inhibitory effect on MCF-7 cells with IC_50_ > 20 μM with a high selective index (SI > 4 of IC_50_ on MDA-MB-468/IC_50_ on MCF-7), implying that **5** displayed strong selective antiproliferative potency against TNBC cells. Other compounds (**2**–**4**, **6**–**8**) retaining α-methylene moiety exhibited weak inhibitory effects against three BrCa cells. The result hinted the importance of the skeleton functionality of **5** but not only the α-methylene moiety in terms of anti-TNBC potency, which will be revealed in the future.

## 3. Materials and Methods

### 3.1. General

An Autopol III automatic polarimeter (Rudolph Research Analytical, NJ, USA) was used to measure the optical rotations. A Bruker Av-400 NMR spectrometer (Bruker BioSpin, Switzerland) was used to record the NMR spectra, with tetramethylsilane (TMS) as an internal standard at room temperature. Column chromatography (CC) was carried out using RP-18 gel (ODS-AQ-HGGEL, AQG12S50, YMC, Co., Ltd., Kyoto, Japan), silica gel (200–300 mesh, Qingdao Marine Chemical Industrials, China) and Sephadex LH-20 (GE Healthcare, Inc., Uppsala, Sweden). Thin-layer chromatography (TLC) (Huanghai Marine Chemical, Ltd., Qingdao, China) was used to monitor the fractions. Agilent 1200 equipment with an Inert Sustain C18 column (5 μm particle size, 5 mm × 250 mm) was used to conduct HPLC analysis. Preparative HPLC was conducted on an NP7005C series instrument using a YMC-Pack ODS-A (10 × 250 mm).

### 3.2. Plant Materials

The *I. britannica* flowers were collected in October 2017 from the river-bed region of Qinling Moutain in Baoji City of Shaanxi province, China. Maintained dry and ventilated, the flowers were identified by Prof. Jun-Mian Tian. A voucher specimen (TJJ-201700119) was kept in a cool and dry environment of Shaanxi Key Laboratory of Natural Products & Chemical Biology, Northwest A&F University.

### 3.3. Extraction and Isolation

We used 95% EtOH (3 × 200 L) to extract the dried flowers of *I. britannica* L. (50 kg) for 12 h at reflux. The combined extracts were then concentrated under reduced pressure to obtain the remaining extract (1.8 kg). After that, the mixture was dissolved in water (20 L) and partitioned with petroleum ether (3 × 20 L), CH_2_Cl_2_ (3 × 20 L) and ethyl acetate (3 × 20 L) in that order. To obtain fourteen fractions (F_1_–F_14_), the EtOAc-soluble fraction (360 g) was chromatographed on a silica gel column eluting with a CH_2_Cl_2_/MeOH (100:0–1:1 *v*/*v*) gradient. Fraction F_3_ (64 g) was subjected to medium-pressure liquid chromatography (MPLC) over octadecylsilica (ODS) with a stepwise gradient of MeOH-H_2_O (from 10% to 100%) to afford nine subfractions (F_3-1_–F_3-9_). Subfraction F_3-3_ (5.6 g) was fractioned on silica gel CC with mixtures of CH_2_Cl_2_/MeOH (100:1-20:1 *v*/*v*) as eluents gradient to obtain six fractions (F_3-3-1_–F_3-3-6_). The further purification of F_3-3-4_ by preparative HPLC (35% CH_3_CN in water) afforded compound **3** (t_R_ = 25 min, 1.6 mg). Subfraction F_3-2_ (5.6 g) was subjected to silica gel column chromatography (CC) with mixtures of DCM-MeOH (80:1-10:1 *v*/*v*) as eluents gradient to obtain ten fractions (F_3-2-1_–F_3-2-10_). Compound **2** (130.6 mg) was isolated after CC over Sephadex LH-20 from F_3-2-4_. By the same procedures, compound **4** (6.7 mg) was obtained from F_3-2-2_. Six fractions (F_3-5-1_–F_3-5-6_) were obtained by fractionating subfraction F_3-5_ on silica gel CC using a gradient of PE-CH_3_COCH_3_ (8:1-1:1 *v*/*v*) mixtures as eluents. Compound **5** (t_R_ = 40 min, 3.6 mg) was obtained after further purification of F_3-5-2_ using preparative HPLC (50% CH_3_CN in water). Similarly, subfraction F_3-6_ (3.9 g) was purified by a Sephadex LH-20 column (eluted with CH_2_Cl_2_/MeOH, *v*/*v*, 1:1) and then a silica gel column eluting with mixtures of PE-EtOAc (8:1-1:2 *v*/*v*) to yield four fractions (F_3-6-4_–F_3-6-4_). Compound **6** was then separated from F_3-6-4-4_ by preparative TLC. Subfraction F_3-7_ (3.2 g) was purified by a Sephadex LH-20 column (eluted with CH_2_Cl_2_: MeOH, *v*/*v*, 1:1) to yield four fractions (F_3-7-1_–F_3-7-4_), and compounds **7** (t_R_ = 25 min, 2.3 mg) and **8** (t_R_ = 27 min, 13.9 mg) were isolated from F_3-7-3_ by preparative HPLC (50% CH_3_CN in water). Fraction F_5_ (64 g) was subjected to MPLC over ODS with a stepwise gradient of MeOH-H_2_O (from 10% to 100%) to afford nine subfractions (F_5-1_–F_5-9_). Subfraction F_5-7_ (2.0 g) was fractioned on silica gel CC with mixtures of PE-EtOAc (8:1-1:2 *v*/*v*) as eluents gradient to obtain twenty fractions (F_5-7-1_–F_5-7-20_). Compound **1** (1.3 mg) was isolated after CC over Sephadex LH-20 from F_5-7-14_.

### 3.4. X-ray Crystallographic Analysis of ***4***

Crystal Data for C_15_H_20_O_4_ (M =264.31 g/mol): orthorhombic, space group P2_1_2_1_2_1_, *a* = 9.55914(9) Å, *b* = 10.00802(10) Å, *c* = 13.83442(15) Å, *V* = 1323.51(2) Å^3^, *Z* = 4, *T* = 149.96(10) K, μ(Cu Kα) = 0.779 mm^−1^, *Dcalc* = 1.326 g/cm^3^, 7356 reflections measured (10.912° ≤ 2Θ ≤ 154.8°), 2626 unique (*R*_int_ = 0.0173, *R*_sigma_ = 0.0153), which were used in all calculations. The final *R*_1_ was 0.0301 (I > 2σ(I)) and *wR*_2_ was 0.1044 (all data).

### 3.5. Cell Culture

MDA-MB-231, MDA-MB-468 and MCF-7 breast cancer cells were seeded in T75 flasks at 2 × 10^6^ cells/flask in high-glucose Dulbecco’s modified Eagle’s medium (DMEM) containing 10% fetal bovine serum (FBS), a 1% penicillin/streptomycin mix and were incubated at 37 °C in an atmosphere of 5% CO_2_. The medium was renewed twice a week, and cells were passaged once a week at a subcultivation ratio of 1:3.

### 3.6. SRB Assay

This SRB method was based on SRB’s ability to stoichiometrically bind to proteins under mildly acidic conditions before being extracted under basic conditions. As a result, the amount of bound dye could approximately represent the cell mass, from which we could extrapolate to measure cell proliferation [28]. MDA-MB-231, MDA-MB-468 and MCF-7 breast cancer cells with the densities of 1.2 × 10^4^, 8 × 10^3^, and 8 × 10^3^ (cells/well) were inoculated into 96-well plates for 24 h, then, the medium was replaced with a fresh one containing the specified compound. Eight compounds were configured into different media, and each compound was set with five different concentrations of 0.1, 0.5, 1, 5, 10, 20 μM. After 48 h of incubation, cell monolayers were fixed with 10% (*w*/*v*) trichloroacetic acid for 12 h, washed with distilled water six times, and stained by SRB (0.4%, *w*/*v*) for 30 min. Then, using 1% (*v*/*v*) acetic acid, the excess dye was removed from the cells by washing them repeatedly six times. The protein-bound dye was dissolved in 100 μL Tris base solution for optical density (OD) determination at 560 nm using a microplate reader.

### 3.7. Statistical Analysis

All tests were performed at least in triplicate. The data are displayed as the mean ± standard deviation (SD). The exhibited data were investigated by one-way analysis of variance (ANOVA) using Graph prism 8.0.

## 4. Conclusions

The phytochemical composition and antiproliferative effects of the sesquiterpenoid-enriched *I. britannica* flowers extract were reported. Eight sesquiterpenoid isolates (**1**–**8**), including one dimer and seven monomers, were isolated from *I. britannica* flowers. Importantly, all isolates were evaluated in vitro for their anti-TNBC properties. Specifically, **5** displayed strong antiproliferative potency against TNBC cells with high selective index (SI) on MCF-7 cells (SI > 4 of IC_50_ on MDA-MB-468/IC_50_ on MCF-7). Additionally, all of the monomers (**2**–**8**) (IC_50_–20 μM) showed weaker activities than that of the dimer (**1**) (IC_50_ = 8.82 ± 0.85 μM) on MCF-7 breast cancer cells. To our best knowledge, compound **5** is the first sesquiterpenoid targeting TNBC cells, which provides enlightenment for further research and development of these bioactive compounds as anti-TNBC agents. 

## Figures and Tables

**Figure 1 molecules-27-05230-f001:**
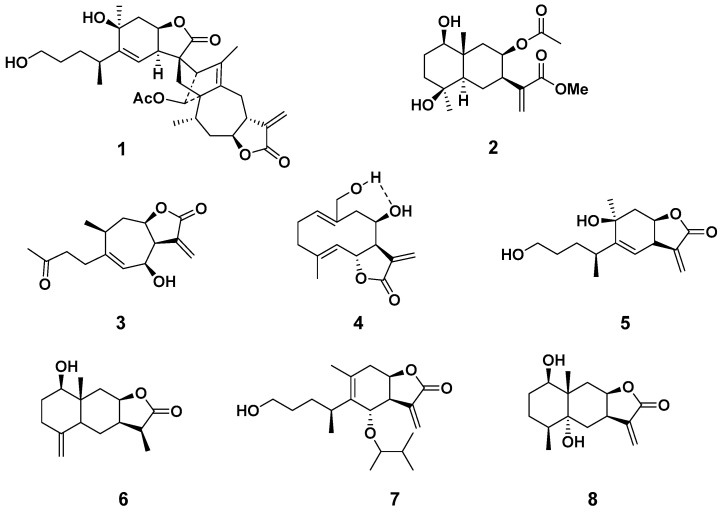
Chemical structure of sesquiterpenoids **1**–**8**.

**Figure 2 molecules-27-05230-f002:**
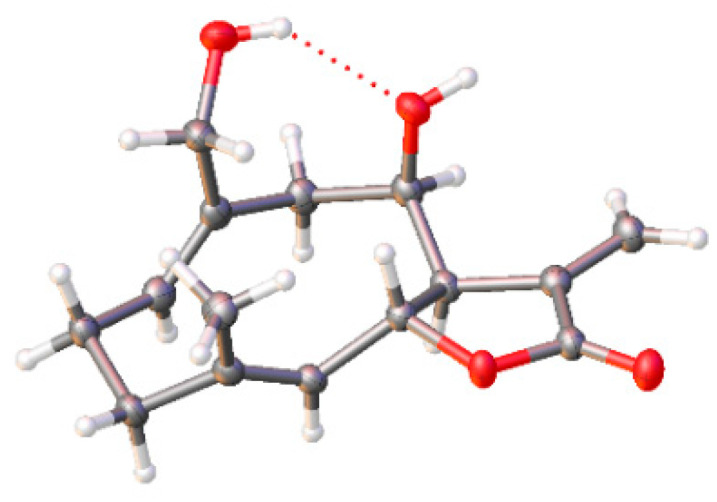
X-ray crystal structure of **4** (Cu Kα radiation, flack parameter: 0.04 (**5**)). Grey balls are carbon (C), white balls are hydrogen (H), and red balls are oxygen (O).

**Table 1 molecules-27-05230-t001:** Antiproliferative activities on breast cancer cells MDA-MB-231, MDA-MB-468 and MCF-7.

Compound	IC_50_ ^1^ (μM)		
	MDA-MB-231	MDA-MB-468	MCF-7
**1**	>20	6.68 ± 0.70	8.82 ± 0.85
**2**	>20	>20	>20
**3**	>20	>20	>20
**4**	>20	>20	>20
**5**	11.5 ± 0.7	4.92 ± 0.65	>20
**6**	>20	>20	>20
**7**	>20	>20	>20
**8**	>20	>20	>20
**DOX ^2^**	2.12 ± 0.19	0.53 ± 0.21	0.15 ± 0.21

^1^ IC_50_ is the concentration that causes 50% inhibition of MDA-MB-231, MDA-MB-468 and MCF-7 breast cancer cells’ proliferation. The data were expressed as the means ± standard deviation. ^2^ Positive control: doxorubicin.

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
