# Peer review of "Sesquiterpenoids from Inula britannica and Their Potential Effects against Triple-Negative Breast Cancer Cells"

_molecules, 2022, doi:10.3390/molecules27165230_

Round 1
Reviewer 1 Report
Comments to the Author
Author/s manuscript entitled “Sesquiterpenoids from Inula britannica and Their Potential Effects against Triple Negative Breast Cancer Cells” is written well.
I have given a few suggestions to improve the manuscript for publication. These mistakes should be corrected before publication.
Abstract:
Numerical results must be included.
The abstract should be ended with a very brief conclusion of the current study.
Materials and methods:
The methodology is not very clear, need more details with references.
Only the SBR assay is not enough to determine cell density by measuring cellular protein content.
Results:
Results of each parameter should be written separately and presented in graphical form.
Add graphical representation of Ultraviolet (UV), electronic circular dichroism (ECD), IR spectra, NMR spectra, and (HR-MS) spectra.
How does α-methylene-γ-butyrolactone act as an anti-cancer agent? Write down the possible mechanism of action.
No information is provided about the concentrations of compounds in the result section.
SBR Assay results are not mentioned in the result sections.
Add GraphPad analysis with graphical form.
Discussion:
The whole manuscript needs careful revision. A comparison of results with previous studies should be written here.
Conclusion
How can we conclude that compound 5 has strong anti-proliferative potency against the MCF-7 cell line?
What is the antiproliferative potential of all other compounds against both MDA-cells?
Author Response
1.Numerical results must be included.
Response: Thanks for pointing it out. We added the numerical results in the abstract part.
2.The abstract should be ended with a very brief conclusion of the current study.
Response: Thanks. We revised the abstract and ended with a very brief conclusion of our work.
Materials and methods:
3.The methodology is not very clear, need more details with references.
Response: We have added more details to the methodology.
4.Only the SBR assay is not enough to determine cell density by measuring cellular protein content.
Response: We have already corrected our description of the SRB assay. The SRB assay has been used since its development in 1990 to inexpensively conduct various screening assays to investigate cytotoxicity in cell-based studies.
Results:
5.Results of each parameter should be written separately and presented in graphical form.
Response: Conventional spectra were placed in the supplementary materials
6.Add graphical representation of Ultraviolet (UV), electronic circular dichroism (ECD), IR spectra, NMR spectra, and (HR-MS) spectra.
Response: All spectra of all compounds were placed in the supplementary materials.
7.How does α-methylene-γ-butyrolactone act as an anti-cancer agent? Write down the possible mechanism of action.
Response: We have written a possible mechanism that α-methylene-γ-butyrolactone acts as an anti-cancer agent.
8.No information is provided about the concentrations of compounds in the result section.
Response: We added the concentration description.
9.SBR Assay results are not mentioned in the result sections.
Response: We added a description of the SBR Assay.
10.Add GraphPad analysis with graphical form.
Response: All data in Table 1 were investigated by one-way analysis of variance (ANOVA) using Graph prism 8.0.
Discussion:
11.The whole manuscript needs careful revision. A comparison of results with previous studies should be written here.
Response: We compared our result of structure identification with previous studies in the discussion part. The anti-TNBC activity of compounds is the first report.
Conclusion
12.How can we conclude that compound 5 has strong anti-proliferative potency against the MCF-7 cell line?
Response: Compound 5 showed anti-proliferative effects on MDA-MB-231 and MDA-MB-468 cells with IC50 of 11.5 ± 0.71 μM and 4.92 ± 0.65 μM. But 5 has no significant inhibitory effect on MCF-7 cells with IC50 > 20 μM. So, 5 has strong anti-proliferative potency against TNBC cells.
13.What is the anti-proliferative potential of all other compounds against both MDA-cells?
Response: Cause all other compounds didn’t show anti-proliferative potential against both MDA-cells (IC50 >20 μM), so we didn’t deeply discuss them in the conclusion part.
Reviewer 2 Report
Comments:
In this manuscript, the authors described “Sesquiterpenoids from Inula britannica and their potential effects against triple negative breast cancer cells”. This paper show that one sesquiterpenoid dimer and seven sesquiterpenoid monomers were isolated from Inula britannica against anti-TNBC. However, there are a few points that need to be clarified.
Comment
1. Compound 1 showed anti-proliferative activities against MDA-MB-468 and MCF-7 cells, but no inhibitory effect on MDA-MB-231. The author shall be more described it in the discussion.
2. Compound 5 has no significant inhibitory effect on MCF-7 cells with IC50 > 20 µM with high selective index. The author shall be more described it in the discussion.
Author Response
- Compound 1 showed anti-proliferative activities against MDA-MB-468 and MCF-7 cells, but no inhibitory effect on MDA-MB-231. The author shall be more described it in the discussion.
Response: Thanks for your advice. We added more details about the anti-proliferative activities of compound 1 against MDA-MB-231 cells.
- Compound 5 has no significant inhibitory effect on MCF-7 cells with IC50 > 20 µM with high selective index. The author shall be more described it in the discussion.
Response: We added more details about the inhibitory effect of Compound 5 on MCF-7 cells.
Reviewer 3 Report
It is an interesting study and contributes sufficiently to overall scientific knowledge. The authors showed that secondary metabolites (sesquiterpenoids) extracted from Inula britannica flowers have the potential against triple-negative breast cancer cells. Compound 5 in particular showed the greatest antiproliferative activity on MDA-MB-231 and MDA-MB-468 breast cancer cells. The strength of this manuscript lies in the new discoveries to elucidate the structure of one sesquiterpenoid extracted from plant materials. Particular attention can be paid to single crystal X-ray diffraction, an advanced method rarely used for the study of plant secondary metabolites.
The manuscript is clear, well organized, and well written. In addition, well-documented test results are collected in Supplementary Materials. Advanced instrumental techniques such as single crystal X-ray diffraction and 1H, 13C NMR, HSQC, HMBC and 1H-1H COSY were used to study the structure of phytochemicals extracted from Inula britannica flowers.
Additionally, with the exception of references 23 and 27, the remaining cited articles were published between 2002 and 2021, mainly in relevant journals.
The drawback of this work is the lack of crystallographic studies for the remaining sesquiterpenoids extracted from Inula britannica flowers.
I have no other criticisms of this manuscript. I think it's a good job.
Author Response
1.The drawback of this work is the lack of crystallographic studies for the remaining sesquiterpenoids extracted from Inula britannica flowers.
Response: Thanks. Most sesquiterpenoids have been reported before, and not all compounds can be precipitated as single crystals. Here, we didn’t do so much of the crystallographic studies.